# TASK-RELEVANT ADVERSARIAL IMITATION LEARNING

## ABSTRACT

We show that a critical problem in adversarial imitation from high-dimensional sensory data is the tendency of discriminator networks to distinguish agent and expert behaviour using task-irrelevant features beyond the control of the agent. We analyze this problem in detail and propose a solution as well as several baselines that outperform standard Generative Adversarial Imitation Learning (GAIL). Our proposed solution, Task-Relevant Adversarial Imitation Learning (TRAIL), uses a constrained optimization objective to overcome task-irrelevant features. Comprehensive experiments show that TRAIL can solve challenging manipulation tasks from pixels by imitating human operators, where other agents such as behaviour cloning (BC), standard GAIL, improved GAIL variants including our newly proposed baselines, and Deterministic Policy Gradients from Demonstrations (DPGfD) fail to find solutions, even when the other agents have access to task reward.

## 1 INTRODUCTION

Generative Adversarial Networks (GANs) have produced breath-taking conditional image synthesis results (Goodfellow et al., 2014; Brock et al., 2019), and have inspired adversarial learning approaches to imitating behavior. In Generative Adversarial Imitation Learning (GAIL) (Ho & Ermon, 2016), a discriminator network is trained to distinguish agent and expert behaviour through its observations, and is then used as a reward function. GAIL agents can overcome the exploration challenge by taking advantage of expert demonstrations, while also achieving high asymptotic performance by learning from agent experience.

Despite the huge promise of GAIL, it has not yet had the same impact as GANs; in particular, robust GAIL from pixels for control applications remains a challenge. Here, we study a key shortcoming of GAIL: the tendency of the discriminator to mainly exploit task-irrelevant features. For example, by focusing on slight background differences, a discriminator can achieve perfect generalization, assigning zero reward to all held-out agent observations. However, this discriminator does not yield an informative reward function because it ignores behavior.

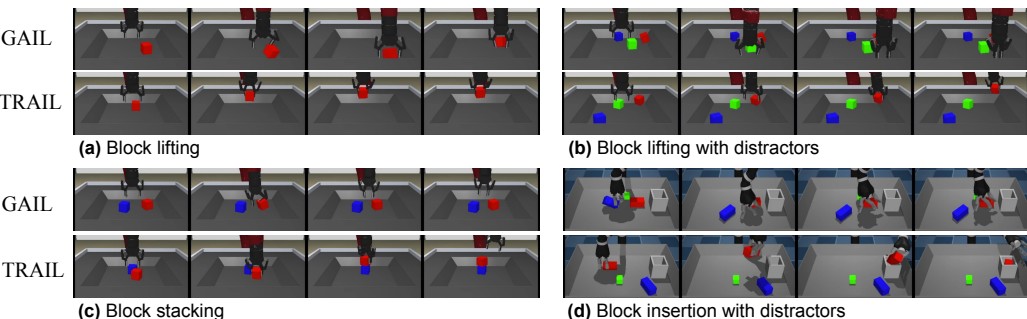

**(a) Block lifting**     **(b) Block lifting with distractors**

**(c) Block stacking**     **(d) Block insertion with distractors**

Figure 1: GAIL and TRAIL succeed at lifting (a), but when distractor objects are added, GAIL fails while TRAIL succeeds (b). Due to robustness to initial conditions, TRAIL can stack from pixels while standard GAIL fails (c). We witness this difference again in insertion with distractors (d). A video showing agents performing these tasks can be seen at `https://youtu.be/Rz5G15rDKcg`.

Changing distractor props    Changing agent appearance    Changing object appearance

Expert Demo 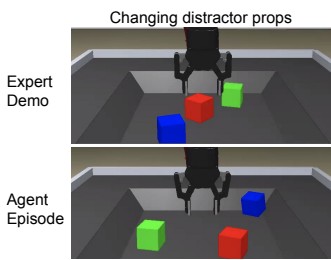 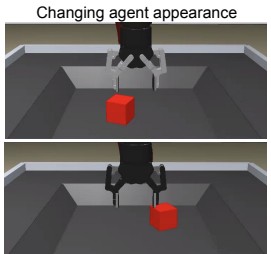 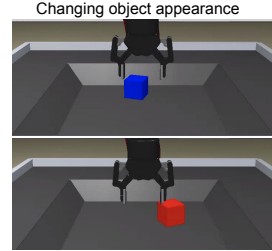

Agent Episode

Figure 2: Illustration of several task-irrelevant changes between the expert demonstrations and the distribution of agent observations, for the *lift* (red cube) task. The naively-trained discriminator network will use these differences rather than task performance to distinguish agent and expert.

Assuming there is an expert policy $\pi_E$ that is optimal for an unknown reward function, here we refer to a feature as *task-irrelevant* if it does not affect that reward. For example, if the task is to lift a red block, the positions of other blocks would be task-irrelevant; see Figures 1 and 2.

This paper makes the following contributions:

1. It reveals a fundamental limitation of GAIL by showing that discriminators do in practice exploit task-irrelevant information, thereby resulting in poor task performance.

2. It introduces powerful GAIL baselines. In particular, it shows that standard regularization and data augmentation are generally useful and improve upon standard GAIL.

3. It shows that these improvements to GAIL, as well as other improvements proposed by Reed et al. (2018), do not completely solve the problem, allowing GAIL agents to fail catastrophically with the addition of task-irrelevant distractors.

4. It introduces *Task Relevant Adversarial Imitation Learning (TRAIL)*, using constrained optimization to force the discriminator to focus on the relevant aspects of the task, which improves performance dramatically on manipulation tasks from pixels (see Figure 1).

## 2 RELATED WORK

The use of demonstrations to help agent training has been studied extensively in robotics (Bakker & Kuniyoshi, 1996; Kawato et al., 1994; Miyamoto et al., 1996) with approaches ranging from Q-learning (Schaal, 1997) to behavioral cloning (BC) (Pomerleau, 1989).

**BC**: BC is effective in solving many control problems (Pomerleau, 1989; Finn et al., 2017; Duan et al., 2017; Rahmatizadeh et al., 2018). It has also been successfully applied to initialize RL training (Rajeswaran et al., 2017). It, however, suffers from compounding errors as initial small deviations from the expert behaviors tend to cause bigger differences (Ross et al., 2011). This often necessitates a large number of demonstrations for satisfactory performance. Furthermore, BC typically does not lead to agents that are superior to their demonstrators.

**Inverse RL**: Ziebart et al. (2008); Ng et al. (2000); Abbeel & Ng (2004) propose inverse reinforcement learning (IRL) as a way of learning reward functions from demonstrations. Reinforcement learning can then be used to optimize that learned reward. Recently, Finn et al. (2016b) approached continuous robotic control problems with success by applying Maximum Entropy IRL algorithms which are very closely related to GAIL (Finn et al., 2016a) and have similar drawbacks.

**Learning from Demonstrations**: Hester et al. (2018) developed deep Q-Learning from demonstration (DQfD), in which expert trajectories are added to experience replay and jointly used to train agents along with their own experiences. This was later extended by Vecerik et al. (2017) and Pohlen et al. (2018) to better handle sparse-reward problems in control and Atari games respectively. Despite their efficiency, this class of methods still requires access to rewards in order to learn.

**GAIL**: Following the success of Generative Adversarial Networks (Goodfellow et al., 2014) in image generation, GAIL (Ho & Ermon, 2016) applies adversarial learning to the problem of imitation. Although many variants are introduced in the literature (Li et al., 2017; Fu et al., 2018; Merel et al., 2017; Zhu et al., 2018; Baram et al., 2017), making GAIL work for high-dimensional input spaces, particularly raw pixels, remains a challenge.

A few papers (Peng et al., 2018; Reed et al., 2018; Blondé & Kalousis, 2018) seek to address the problem of overfitting the discriminator. Peng et al. (2018) introduces the Variational Bottleneck to regularize the discriminator. Reed et al. (2018) proposes to not train the vision module of the discriminator (e.g. use the vision module of the critic network instead) and only train a tiny network on top of the vision module to discriminate. Blondé & Kalousis (2018) follow a similar approach. Unstructured regularization, however, cannot stop the discriminator from fitting to features that are systematically different between the agents' and the demonstration's behavior, like those illustrated in Figure 2. TRAIL, on the other hand, is much less prone to overfitting to these features.

Stadie et al. (2017) extend GAIL to the setting of third person imitation, in which the demonstrator and agent observations come from different views. To prevent the discriminator from discriminating based on viewpoint domain, they use gradient flipping from an auxiliary classifier to learn domain-invariant features. Our approach is not to learn domain-invariant features, but instead learn domain-agnostic discriminators that only focus on behavior.

Several recent works have focused on improving the sample efficiency of GAIL (Blondé & Kalousis, 2018; Sasaki et al., 2018). Common to these approaches and to this work, is the use of off-policy actor critic agents and experience replay, to improve the utilization of available experience.

## 3 REINFORCEMENT LEARNING AND ADVERSARIAL IMITATION

Following the notation of Sutton & Barto (2018), a Markov Decision Process (MDP) is a tuple $(\mathcal{S}, \mathcal{A}, R, P, \gamma)$ with states $\mathcal{S}$, actions $\mathcal{A}$, reward function $R(s, a)$, transition distribution $P(s'|s, a)$, and discount $\gamma$. An agent in state $s \in S$ takes action $a \in A$ according to its policy $\pi$ and moves to state $s' \in \mathcal{S}$ according to the transition distribution. The goal of RL algorithms is to find a policy that maximizes the expected sum of discounted rewards, represented by the action value function $Q^\pi(s, a) = \mathbb{E}^\pi[\sum_{t=0}^\infty \gamma^t R(s_t, a_t)]$, where $\mathbb{E}^\pi$ is an expectation over trajectories starting from $s_0 = s$ and taking action $a_0 = a$ and thereafter running the policy $\pi$.

To apply RL, it is essential that we have access to the reward function which is often hard to design and evaluate (Singh et al., 2019). In addition, sparse rewards can cause exploration difficulties that pose great challenges to RL algorithms. We therefore look to imitation learning and particularly GAIL to derive a reward function from expert demonstrations. In GAIL, a reward function is learned by training a discriminator network $D(s, a)$ to distinguish between agent and expert state-action pairs. The GAIL objective is thus formulated as follows:

$$\min_\pi \ \max_D \ \mathbb{E}_{(s,a)\sim\pi_E}[\log D(s, a)] + \mathbb{E}_{(s,a)\sim\pi}[\log(1 - D(s, a))] - \lambda_H H(\pi), \qquad (1)$$

where $\pi$ is the agent policy, $\pi_E$ the expert policy, and $H(\pi)$ an (optional) entropy regularizer. The reward function is defined simply: $R(s, a) = -\log(1 - D(s, a))$.

GAIL is theoretically appealing and practically simple. The discriminator, however, can focus on any features to discriminate, whether these features are task-relevant or not. In the next subsection we describe a way to constrain the discriminator network in order to prevent it from using task-irrelevant details to distinguish agent and expert data.

## 4 TASK-RELEVANT ADVERSARIAL IMITATION LEARNING (TRAIL)

We want the discriminator to focus on task-relevant features. Our proposed solution, TRAIL, prevents the discriminator from being able to distinguish expert and agent behaviour based on selected aspects of the data. For instance, the discriminator should distinguish agent and expert frames only when meaningful behavior is present in those frames. In the absence of behavior useful to solve the task, *e.g.* in initial frames prior to the execution of the behavior, the discriminator should be agnostic.

To this end, we propose a constraint for the discriminator, such that its accuracy must not be greater than chance on a particular set of observations $\mathcal{I}$ that we will call the invariant set. $\mathcal{I}$ will include only observations that can be distinguished as agent or expert in task-irrelevant ways. Precisely, we

formulate TRAIL in terms of the following constrained optimization problem:

$$\max_{\psi} \quad \mathbb{E}_{s \sim \pi_E} \left[ \log D_\psi(s) \right] + \mathbb{E}_{s \sim \pi_\theta} \left[ \log(1 - D_\psi(s)) \right] \tag{2}$$

$$s.t. \quad \frac{1}{2} \mathbb{E}_{s \sim \pi_E} \left[ \mathbf{1}_{\mathrm{D}_\psi(\mathrm{s}) \geq \frac{1}{2}} \mid s \in \mathcal{I} \right] + \frac{1}{2} \mathbb{E}_{s \sim \pi_\theta} \left[ \mathbf{1}_{\mathrm{D}_\psi(\mathrm{s}) < \frac{1}{2}} \mid s \in \mathcal{I} \right] \leq \frac{1}{2}.$$

To apply the above constraint in practice, we can optimize the reverse of the objective function on $\mathcal{I}$. That is, given a batch of $N$ examples $s_e \sim \pi_E$, $s_\theta \sim \pi_\theta$ from the expert and agent, and $\hat{s}_e \sim \pi_E$ and $\hat{s}_\theta \sim \pi_\theta$ both in the set $\mathcal{I}$, we maximize the following augmented objective function:

$$\mathcal{L}_\psi(s_e, s_\theta, \hat{s}_e, \hat{s}_\theta) = \sum_{i=1}^{N} \log D_\psi\left(s_e^{(i)}\right) + \log\left(1 - D_\psi\left(s_\theta^{(i)}\right)\right) \tag{3}$$

$$- \lambda \left[ \sum_{i=1}^{N} \log D_\psi\left(\hat{s}_e^{(i)}\right) + \log\left(1 - D_\psi\left(\hat{s}_\theta^{(i)}\right)\right) \right] \mathbf{1}_{accuracy(\hat{s}_e, \hat{s}_\theta) \geq \frac{1}{2}},$$

where $accuracy(\cdot, \cdot)$ is defined as the average of discriminator accuracies:

$$accuracy(\hat{s}_e, \hat{s}_\theta) = \frac{1}{2N} \sum_{i=1}^{N} \left[ \mathbf{1}_{\mathrm{D}_\psi\left(\hat{s}_e^{(i)}\right) \geq \frac{1}{2}} + \mathbf{1}_{\mathrm{D}_\psi\left(\hat{s}_\theta^{(i)}\right) < \frac{1}{2}} \right]. \tag{4}$$

The scalar $\lambda \geq 0$ is a tunable hyperparameter.

We used single frames as states but sequences can also be used, and we do not require expert actions. Not requiring actions enables learning in very off-policy settings, where the action dimensions and distributions of the demonstrator (another robot or human) are different from those of the agent.

### 4.1 THE SELECTION OF THE INVARIANT SET

The selection of the invariant set $\mathcal{I}$ is a design choice. In general, we could always contrive non-stationary and adversarial ways of making this choice difficult. However, we argue that in many situations of great interest, including our robotic manipulation setup, it is easy to propose effective and very general invariant sets.

A straightforward way to collect robot data is to execute a *random policy*. We can then use the resulting random episodes, for both expert and agent, to construct the invariant set $\mathcal{I}$. Another way to construct $\mathcal{I}$ is to use *early frames* from both expert and agent episodes. Since in early frames little or no task behavior is apparent, this strategy turns out to be effective and no extra data has to be collected. This strategy also improves robustness with respect to variation in the initial conditions of the task; see for example block insertion in Figure 1(d).

Importantly, if the set $\mathcal{I}$ captures some forms of irrelevance but not all forms, it will nonetheless always help in improving performance. In this regard, TRAIL will dominate its GAIL predecessor whenever the designer has some prior on what aspects of the data might be task irrelevant.

## 5 EXPERIMENTS

We focus on solving robot manipulation tasks. The environment implements two work-spaces: one with a Kinova Jaco arm (*Jaco*), and the other with a Sawyer arm (*Sawyer*). See supplementary material A.1 for a detailed description. Environment rewards, which are not used by GAIL-based methods, are sparse and equal to +1 for each step when a given task is solved and 0 otherwise. The maximum reward for the episode is 200, since this is the length of a single evaluation episode.

Our agent is based on the off-policy D4PG algorithm (Barth-Maron et al., 2018) because of its stability and data-efficiency (see supplementary material A.6). Following Vecerik et al. (2017), we add expert demonstrations into the agents' experience replay, and refer to the resulting RL algorithm as D4PG from Demonstrations (D4PGfD). For each task we collect 100 human demonstrations.

**Data augmentation** Traditional data augmentation has proved beneficial in imitation learning (Berseth & Pal, 2019). Surprisingly, to the best of our knowledge, this has not been explicitly studied in prior publications on GAIL. However, we find that data augmentation is a generally useful component to prevent discriminator overfitting. It drastically improves the baseline GAIL agent, and

is *necessary* to solve any of the harder manipulation tasks. We distort images by randomly changing brightness, contrast and saturation; random cropping and rotation; adding Gaussian noise. When multiple sensor inputs are available (e.g. multiple cameras), we also randomly drop out these inputs, but leaving at least one active.

All discriminator-based methods in this work (including all baselines) use data augmentation except as indicated in the relevant ablations (see Section 5.2). We also considered regularizing the GAIL discriminator with spectral normalization (Miyato et al., 2018). It performed slightly better than GAIL, but still failed in the presence of distractor objects, and we thus omit spectral normalization in the main experiments for simplicity.

**Actor early stopping** When the agent has learned the desired behavior, and the resulting data is used for training, the discriminator will become unable to distinguish expert and agent observations based only on behavior. This forces the discriminator to rely on task-irrelevant information.

To avoid this scenario, we propose to restart each actor episode after a certain number of steps such that successful behavior is rarely represented in agent data. This enables the discriminator to recognize the goal condition, which appears frequently at the end of demonstration episodes, as representative of expert behavior. To avoid hand-tuning the stopping step number, we found that the discriminator score can be used to derive an adaptive stopping criterion. Concretely, we restart an episode if the discriminator score at the current step exceeds the median score of the episode so far for $T_{patience}$ consecutive steps (in practice we set $T_{patience} = 10$).

We set invariant set hyperparameter $\lambda = 1$ and use adaptive early stopping for TRAIL. The ablation with $\lambda = 0$, *i.e.* adaptive early stopping only, is referred to as TRAIL-0. Hence, the only difference between TRAIL-0 and TRAIL is that the later uses invariant set constraints.

## 5.1 BLOCK LIFTING WITH DISTRACTORS

In this section, we consider two variants of the *lift* task in the *Sawyer* work space: a) *lift alone*, where only one red cube is present, b) *lift distracted*, with two extra blocks (blue and green, see Figure 9). We show how adding these additional distractors affects the training procedure.

We first compare our method to baselines. In doing so, the invariant set is constructed using the first 10 frames from every episode, as in the rest of the work. This choice does not require us to collect any extra data, and hence the comparison with baselines is fair. In the next section, we elaborate on the choice of the invariant set and provide additional experimental results.

As baselines, we run BC and GAIL (with data augmentation). Baseline GAIL is the strongest we were able to implement. The only difference between this baseline and TRAIL is the use of invariant set constrains and actor early stopping (code, agent configurations like number of actors, and network architecture are the same). We additionally consider the approaches proposed by Reed et al. (2018) as GAIL-based baselines; using either a randomly initialized convolutional network, or a convolutional critic network, to provide fixed vision features on top of which a tiny discriminator network is trained. We call these two baselines *random* and *critic* respectively. Finally, to show the importance of actor early stopping, we run TRAIL-0 (Fig. 3).

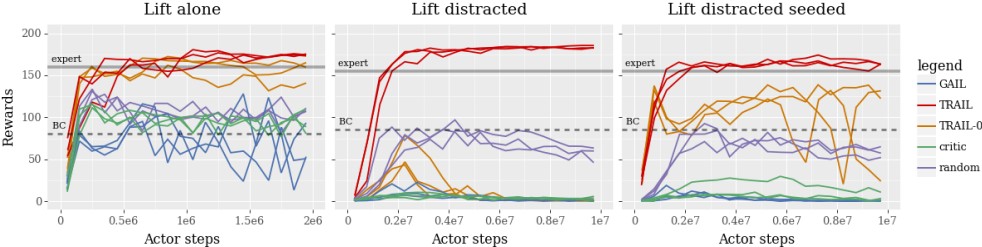

Figure 3: Results for *lift alone*, *lift distracted*, and *lift distracted seeded*. Only TRAIL excels.

All methods perform satisfactorily on *lift alone*, but the proposed methods TRAIL-0 and TRAIL do best. As expected, the performance of BC on *lift distracted* is similar to its performance on *lift alone*, despite the two additional blocks. The two additional blocks in *lift distracted* affect the GAIL-based baselines, despite being irrelevant to the task.

To understand this effect, we conducted an additional experiment (*lift distracted seeded*). Here the initial block positions are randomly drawn from the expert demonstrations. Therefore, it is impossible to discriminate between expert and actor episodes using the first few frames of an episode. Note this initialization procedure is not applied to the evaluation actor, keeping the evaluation scores comparable between *lift distracted* and *lift distracted seeded*.

This experiment exposes one major culprit behind the performance degradation of GAIL: memorization. The discriminator can achieve perfect accuracy by memorizing all 100 initial positions from the demonstration set, making the reward function uninformative. By constraining the discriminator, TRAIL squeezes out this irrelevant information and succeeds in solving the task in the presence of distractions. TRAIL is the only method that is able to handle the variety of initial cube positions during training, achieving better than expert performance on *lift distracted*.

Interestingly, *random* performs reasonably on *lift distracted*. Given *random*'s strong performance, we conducted additional experiments to evaluate its effectiveness when trained with adaptive early stopping and present the results in Figure 12 of the supplementary material.

### CONSTRUCTING THE INVARIANT SET $\mathcal{I}$

In the previous subsection, early frames were used to construct the invariant set (TRAIL-early). Here, we evaluate another previously mentioned approach for constructing $\mathcal{I}$; random policy (TRAIL-random). The *lift distracted* task caused all baselines to fail, but was solved by TRAIL. We introduce a harder version of the task, where the expert appearance is different, to tease out the differences between TRAIL-early and TRAIL-random. The difference in appearance between the expert and imitator allows the GAIL discriminator to trivially distinguish them. The results and the differences in the expert appearance are presented in Figure 4.

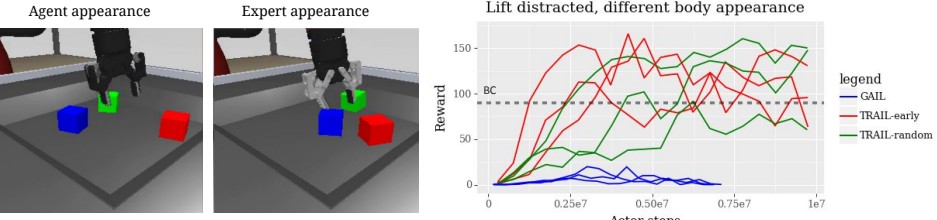

Figure 4: Lift red block, where expert has a different body appearance, and with distractor blocks. TRAIL-random outperforms GAIL, and performs on par with TRAIL-early.

The new task is indeed harder and it takes longer for TRAIL methods to achieve performance better than BC baseline, which is not affected by the different body appearance. GAIL is clearly outperformed and does not take off.

The difference between TRAIL methods is negligible. We also tried to mix them but the differences remain imperceptible. Hence, in the following experiments we simply use early frames. This choice is pragmatic as it does not require that we collect any extra data, and hence the comparison with GAIL and other baselines is fair. It is also very simple to apply in practice, even if one does not have access to the expert setup anymore. Finally, it is general and powerful enough to be successfully used across all robotic manipulation tasks considered in this work.

To decide how many initial frames should be used to construct $\mathcal{I}$, we conducted an ablation study and found out that the method is not very sensitive to this choice (see supplementary material A.2). Hence, we chose 10 initial frames, and intentionally used the same number for all tasks to further emphasize generality of this choice.

## 5.2 ABLATION STUDIES

### MEASURING DISCRIMINATOR MEMORIZATION OF TASK-IRRELEVANT FEATURES

In this section, we experimentally confirm that memorization is a limiting factor of GAIL methods. We equipped the discriminator with two extra heads whose inputs are the final spatial layer of the

ResNet for the *lift distracted* task. The first head is trained on the first frames only, and has the same target as the main head (*i.e.* discriminating between agent and expert). To train the second head, we randomly divide the expert demonstrations into two equi-numerous subsets, and the task is to predict to which of these randomly chosen sets the demonstration was assigned to. Both heads are trained via backpropagation but their gradient is not propagated to the ResNet so they do not influence the training procedure directly.

If our claim is correct, we expect the extra heads to have higher accuracy for TRAIL-0 as compared to TRAIL, since TRAIL representations are penalized for having features triggering memorization, *i.e.* the features should not aid discriminating based on the first frames or predicting a random label for each expert demonstration. No reasonably performing method is able to force 50% accuracy for extra heads since some features are important to solve the task (*e.g.* the position of the red cube).

We also collected 25 extra holdout demonstrations and visualize the average discriminator prediction on them, and compare with predictions on the training demonstrations.

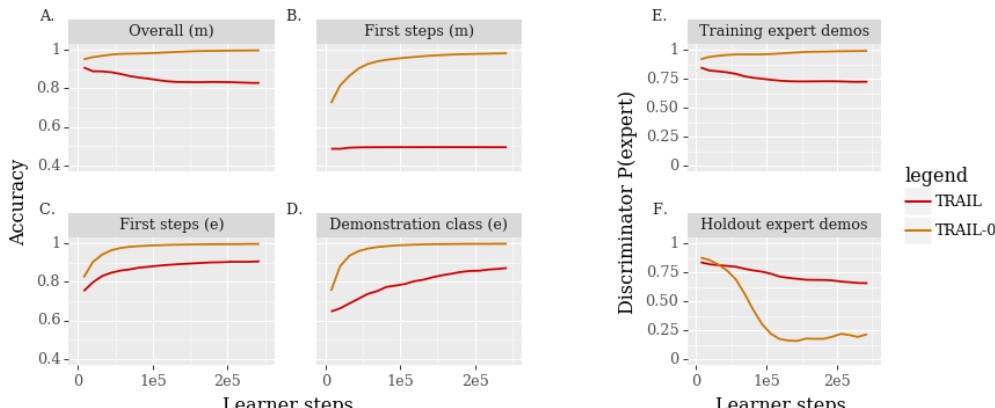

Figure 5: Demonstrating the memorization problem on the *lift distracted* task (here higher accuracy is worse). Accuracy of different discriminator heads is presented (A-D). In A, the overall accuracy for all timesteps. Then main (m) and extra (e) heads accuracy for the first steps are presented in B and C, respectively. Accuracy of the head predicting randomly assigned demonstration class is shown in D. Average discriminator predictions for training and holdout demonstration are shown in E and F.

In Fig. 5, we see the overall accuracy of the main head for TRAIL-0 is significantly higher, and the difference is larger for early steps (when TRAIL achieves only 50%, as expected). TRAIL representations are less helpful for extra heads (see Figure 5 C-D). Finally, TRAIL-0 clearly overfits on training demonstrations, predicting almost the maximum score, while only $\sim 0.25$ is predicted for holdout demonstrations. The TRAIL average predictions for both datasets are almost identical.

ACTOR EARLY STOPPING (TRAIL-0)

In this section, we analyze the importance of adaptive early stopping on 3 tasks in the *Jaco* workspace: lift red cube (*lift*), put red cube in box (*box*), and stack red cube on blue cube (*stack*). We consider D4PGfD and three GAIL-based models with varying termination policies: a) fixed step (50), b) based on ground truth task rewards, and c) based on adaptive early stopping (TRAIL-0). Using ground truth task rewards, an episode is terminated if the reward at the current step exceeds the median reward of the episode so far for 10 consecutive steps. Results are presented in Figure 6.

Termination based on task reward is clearly superior; although unrealistic in practice, it defines the performance upper-bound and clearly shows that early stopping is beneficial. TRAIL-0 is robust and reaches human performance on all tasks. A fixed termination policy, when tuned, can be very effective. The same fixed termination step, however, does not work for all tasks. See Figure 13 in supplementary material for the effects of varying termination steps. Finally, as can be inferred from *stack* results, the dense rewards provided by TRAIL-0 are helpful in solving this challenging problem which is unsolved with D4PGfD even though D4PGfD uses ground truth task rewards.

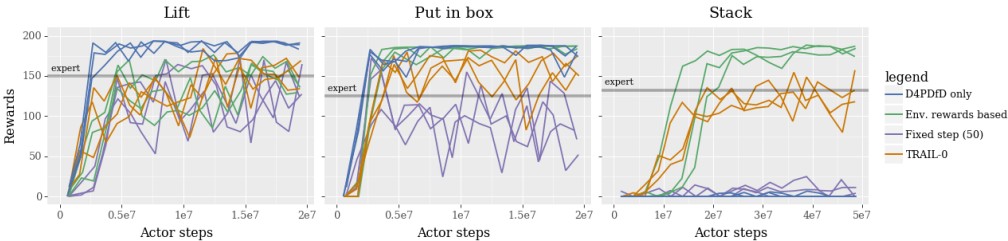

Figure 6: Results for *lift*, *box*, and *stack* on *Jaco* environment.

DATA AUGMENTATION

In Table 5.2, we report the best reward obtained in the first 12 hours of training (averaged for all seeds; see Figure 14 for full curves). The results show that data augmentation is needed in *lift distracted*. For the *lift alone* task, TRAIL-0 with data augmentation performs on par with TRAIL.

| Task | Regularization | Data augmentation | No data augmentation |
|---|---|---|---|
| *lift alone* | TRAIL-0 | $\sim$165 | $\sim$115 |
| | TRAIL | $\sim$155 | $\sim$165 |
| *lift distracted* | TRAIL-0 | $\sim$30 | $\sim$5 |
| | TRAIL | $\sim$180 | $\sim$10 |

Table 1: Influence of data augmentation (evaluated on rewards) for *lift alone* and *lift distracted*.

The performance of TRAIL is not affected by the lack of data augmentation on the *lift alone* task, whereas the performance of TRAIL-0 is.

LEARNING WITH A FIXED, PERFECT DISCRIMINATOR

To assess whether learned discriminators are necessary, we compare TRAIL against agents using a fixed reward function corresponding to $R_{expert} = 1$ and $R_{agent} = 0$ for the *lift alone* and *lift distracted* tasks. This baseline simulates an oracle discriminator with perfect generalization, but which is agnostic to behavior. On the *lift alone* task, agents using this fixed reward achieve roughly half the reward of TRAIL asymptotically, and on *lift distracted* they do not solve the task (average rewards are less than 5). See supplementary Figure 16 for learning curves.

## 5.3 LEARNING FROM OTHER EMBODIMENTS AND PROPS

Since the TRAIL discriminator is trained to ignore task-irrelevant features, it can learn from demonstrations with different embodiments and props. Figure 7 shows that GAIL even with augmentation fails to learn block lifting from a different embodiment, and performs worse when the expert uses a different prop color. TRAIL solves the task and achieves better performance in both cases.

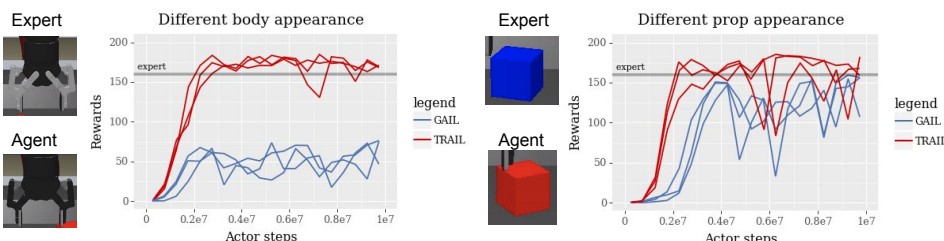

Figure 7: When the expert differs in body or prop appearance, TRAIL outperforms GAIL.

## 5.4 EVALUATION ON DIVERSE MANIPULATION TASKS

To further demonstrate benefits of using our proposed method, we present results for TRAIL, TRAIL-0, the baseline GAIL, and BC on more challenging tasks. Specifically, we consider *stack* with the *Sawyer* robot; and *insertion* and *stack banana* in the *Jaco* work-space. The results are shown in Figure 8. The tasks we consider here are much harder as evidenced by the performance of BC agents. These experiments suggest that TRAIL is generally useful as an improvement over GAIL, even when the tasks are not designed to include task-irrelevant information.

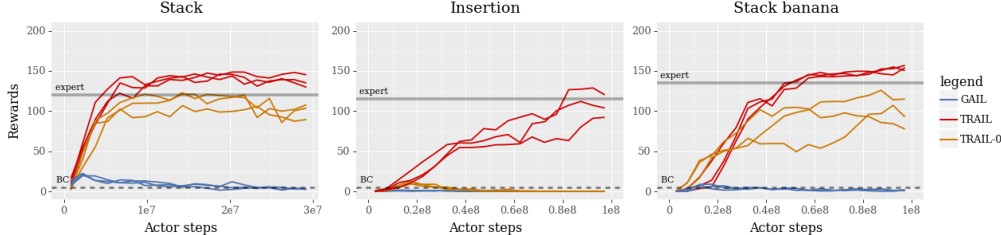

Figure 8: Results comparing TRAIL, TRAIL-0 and GAIL for diverse manipulation tasks.

## 6 CONCLUSIONS

To make adversarial imitation work on nontrivial tasks from pixels, it is crucial to prevent the discriminator from exploiting task-irrelevant information. Data augmentation is a necessary component that prevents the discriminator from overfitting, but alone is not sufficient to solve more challenging problems. Our proposed method TRAIL proved to be a remedy which effectively focuses the discriminator on the task even when task-irrelevant features are present.

We showed through ablations that TRAIL benefits from data augmentation, actor early stopping, and the use of invariant set constraints. The last enables our agent to solve the full task suite in our experiments showing its decisive importance.

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

# A  SUPPLEMENTARY MATERIAL

## A.1  DETAILED DESCRIPTION OF ENVIRONMENT

All our simulations are conducted using MuJoCo[1] (Todorov et al., 2012). We test our proposed algorithms in a variety of different envriomnents using simulated Kinova Jaco[2], and Sawyer robot arms[3]; see Figure 9. We use the Robotiq 2F85 gripper[4] in conjuction with the Sawyer arm.

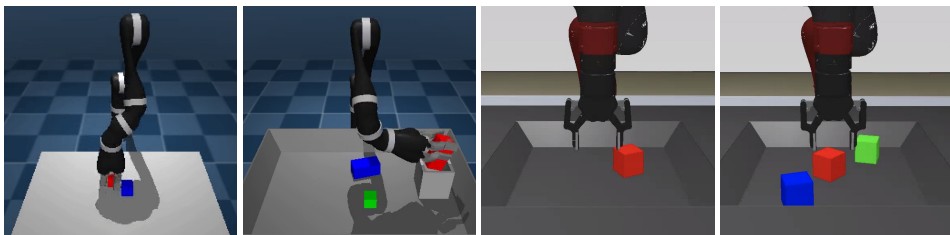

Figure 9: Two work spaces, *Jaco* (left) which uses the Jaco arm and is $20 \times 20$ cm, and *Sawyer* (right) which uses the Sawyer arm and more closely resembles a real robot cage and is $35 \times 35$ cm.

To provide demonstrations, we use the SpaceNavigator 3D motion controller[5] to set Cartesion velocities of the robot arm. The gripper actions are implemented via the buttons on the controller. All demonstrations in our experiments are provided via human teleoperation and we collected 100 demonstrations for each experiment.

**Jaco**: When using the Jaco arm, we use joint velocity control (9DOF) where we control all 6 joints of arm and all 3 joints of the hand. The simulation is run with a numerical time step of 10 milliseconds, integrating 5 steps, to get a control frequency of 20HZ. The agent uses a frontal camera of size $64 \times 64$ (see Figure 10(a)). For a full list of observations the agent sees, please refer to Table 2(a).

(a) FRONTAL CAMERA     (b) FRONT LEFT CAMERA     (c) FRONT RIGHT CAMERA

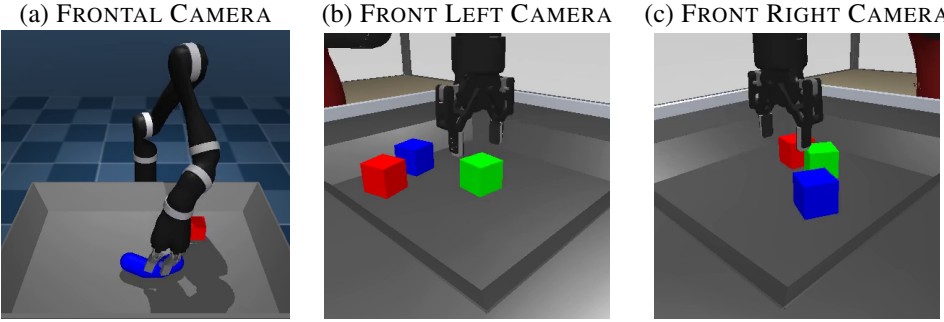

Figure 10: Illustration of the pixels inputs to the agent.

**Sawyer**: When using the Sawyer arm, we use Cartesian velocity control (6DOF) for the robot arm and add one additional action for the gripper resulting in 7 degrees of freedom. The simulation is run with a numerical time step of 10 milliseconds, integrating 10 steps, to get a control frequency of 10HZ. The agent uses two frontal cameras of size $64 \times 64$ situated on the left and right side of the robot cage respectively (see Figure 10(b, c)). For a full list of observations the agent sees, please refer to Table 2(b).

For all environments considered in this paper, we provide sparse rewards (i.e. if task is accomplished, the reward is 1 and 0 otherwise). In experiments regards our proposed methods, rewards are only used for evaluation purposes and not for training the agent.

---

[1] www.mujoco.org
[2] https://www.kinovarobotics.com/en/products/assistive-technologies/
kinova-jaco-assistive-robotic-arm
[3] https://www.rethinkrobotics.com/sawyer/
[4] https://robotiq.com/products/2f85-140-adaptive-robot-gripper
[5] https://www.3dconnexion.com/spacemouse_compact/en/

Table 2: Observation and dimensions.

| a) Jaco | | b) Sawyer | |
|---|---|---|---|
| **Feature Name** | **Dimensions** | **Feature Name** | **Dimensions** |
| frontal camera | $64 \times 64 \times 3$ | front left camera | $64 \times 64 \times 3$ |
| base force and torque sensors | 6 | front right camera | $64 \times 64 \times 3$ |
| arm joints position | 6 | arm joint position | 7 |
| arm joints velocity | 6 | arm joint velocity | 7 |
| wrist force and torque sensors | 6 | wrist force sensor | 3 |
| hand finger joints position | 3 | wrist torque sensor | 3 |
| hand finger joints velocity | 3 | hand grasp sensor | 1 |
| hand fingertip sensors | 3 | hand joint position | 1 |
| grip site position | 3 | tool center point cartesian orientation | 9 |
| pinch site position | 3 | tool center point cartesian position | 3 |
| | | hand joint velocity | 1 |

## A.2 EARLY FRAMES ABLATION STUDY

Here, we vary the number of early frames of each episode used to form $\mathcal{I}$. We report that a range of values from 1 up to 20 works well across both tasks (Put in box and Stack banana), with performance gradually degrading as the value increased beyond 20 (Fig. 11).

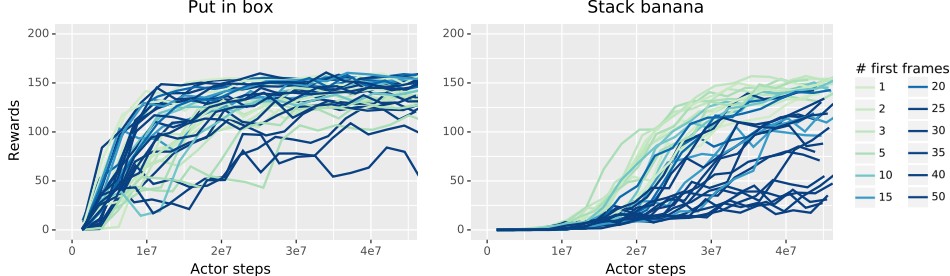

Figure 11: TRAIL performance, varying the number of first frames in each episode used to form $\mathcal{I}$.

## A.3 TRAIL-0 WITH *random*

We found out and mentioned in the subsection 5.1 that *random* benefits from TRAIL-0. Unfortunately, it is still prone to overfitting and hence, worse than our full method – TRAIL. We present *random* + TRAIL-0 accompanied with our methods in Figure 12. Our TRAIL and *random* + TRAIL-0 are the only methods exceeding BC performance on *lift distracted*. However, TRAIL performance is clearly better (obtains higher rewards and never gets overfitted).

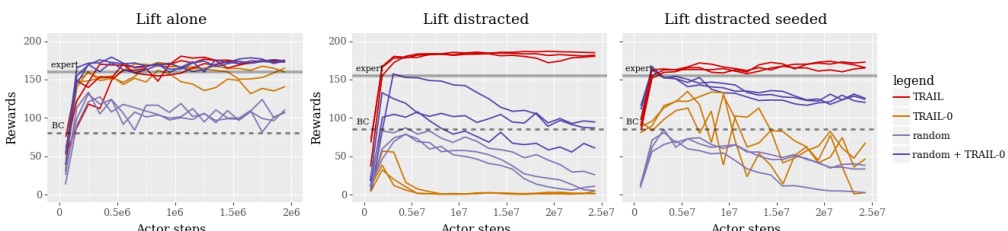

Figure 12: Results for *lift alone*, *lift distracted*, and *lift distracted seeded*.

## A.4 FIXED TERMINATION POLICY

As mentioned in the subsection 5.2, the most basic early termination policy – fixed step termination – may be very effective if tuned. Since the tuning may be expensive in practice, we recommend using adaptive early stopping (TRAIL-0). However, for the sake of completeness we provide results for fixed step termination policy depending on the hyperparameter tuned. The results for *stack* task are

presented in Figure 13. The *Jaco* work space is considered here because *Sawyer* requires TRAIL to obtain high rewards. As can be inferred from the figure, the performance is very sensitive to the fixed step hyperparameter. We refer to subsection 5.2 for more comments on all methods.

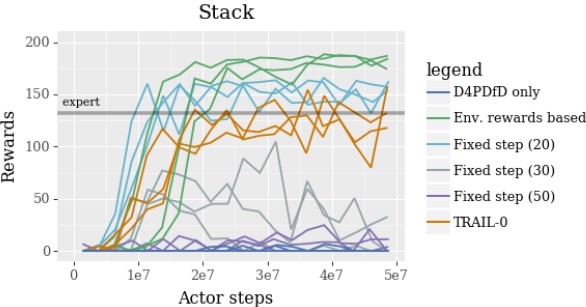

Figure 13: Results for *stack* in *Jaco* work space. Fixed step termination policy can be very effective but the final performance is very sensitive to the hyperparameter. TRAIL-0 does not need tuning nor access to the environment reward.

## A.5 DATA AUGMENTATION

An extra set of experiments on *lift alone* and *lift distracted* tasks (described in subsection 5.1) has been performed to show importance of data augmentation.

Because in the subsection 5.2 only the peak performance is presented (Table 5.2), we present here the full curves in Figure 14. The results shows that data augmentation is necessary to obtain high rewards in *lift distracted*. For easier *lift alone*, TRAIL-0 with data augmentation perform at par with TRAIL.

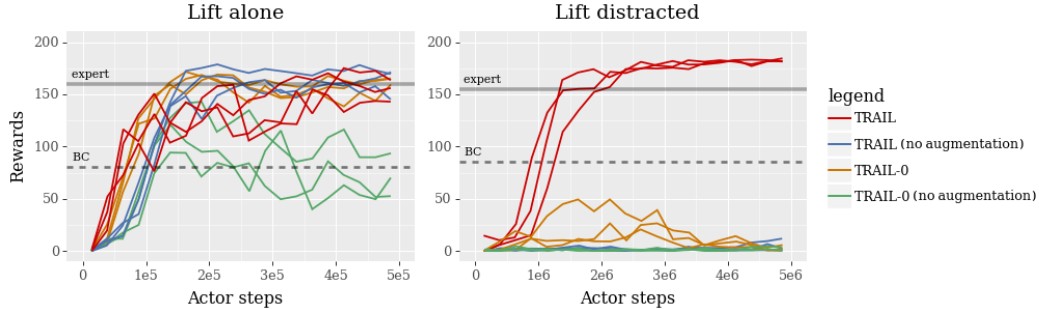

Figure 14: Results for *lift alone* and *lift distracted* in *Sawyer* work space. TRAIL and TRAIL-0 are by default with data augmentation. Additional results for these methods without data augmentation are presented to show its importance.

## A.6 D4PG

We use D4PG (Barth-Maron et al., 2018) as our main training algorithm. Briefly, D4PG is a distributed off-policy reinforcement learning algorithm for continuous control problems. In a nutshell, D4PG uses Q-learning for policy evaluation and Deterministic Policy Gradients (DPG) (Silver et al., 2014) for policy optimization. An important characteristic of D4PG is that it maintains a replay memory $\mathcal{M}$ (possibility prioritized (Horgan et al., 2018)) that stores SARS tuples which allows for off-policy learning. D4PG also adopts target networks for increased training stability. In addition to these principles, D4PG utilized distributed training, distributional value functions, and multi-step returns to further increase efficiency and stability. In this section, we explain the different ingredients of D4PG.

D4PG maintains an online value network $Q(s, a|\theta)$ and an online policy network $\pi(s|\phi)$. The target networks are of the same structures as the value and policy network, but are parameterized by different parameters $\theta'$ and $\phi'$ which are periodically updated to the current parameters of the online networks.

Given the $Q$ function, we can update the policy using DPG:

$$\mathcal{J}(\phi) = \mathbb{E}_{s_t \sim \mathcal{M}}\big[\nabla_\phi Q(s_t, \pi(s_t|\phi)|\theta)\big]. \tag{5}$$

Instead of using a scalar $Q$ function, D4PG adopts a distributional value function such that $Q(s_t, a|\theta) = \mathbb{E}\big[Z(s_t, a|\theta)\big]$ where $Z$ is a random variable such that $Z = z_i$ w.p. $p_i \propto \exp(\omega(s_t, a|\theta))$. The $z_i$'s take on $V_{bins}$ discrete values that ranges uniformly between $V_{min}$ and $V_{max}$ such that $z_i = V_{min} + i\frac{V_{max}-V_{min}}{V_{bins}}$ for $i \in \{0, \cdots, V_{bins} - 1\}$.

To construct a bootstrap target, D4PG uses N-step returns. Given a sampled tuple from the replay memory: $s_t, a_t, \{r_t, r_{t+1}, \cdots, r_{t+N-1}\}, s_{t+N}$, we construct a new random variable $Z'$ such that $Z' = z_i + \sum_{n=0}^{N-1} \gamma^n r_{t+n}$ w.p. $p_i \propto \exp(\omega(s_{t+N}, \pi(s_{t+N}|\phi')|\theta'))$. Notice, $Z'$ no longer has the same support. We therefore adopt the same projection $\Phi$ employed by Bellemare et al. (2017). The training loss for the value function

$$\mathcal{L}(\theta) = \mathbb{E}_{s_t, a_t, \{r_t, \cdots, r_{t+N-1}\}, s_{t+N} \sim \mathcal{M}}\big[H\big(\Phi(Z'), Z(s_t, a_t|\theta)\big)\big], \tag{6}$$

where $H$ is the cross entropy.

D4PG is also distributed following Horgan et al. (2018). Since all learning processes only rely on the replay memory, we can easily decouple the 'actors' from the 'learners'. D4PG therefore uses a large number of independent actor processes which act in the environment and write data to a central replay memory process. The learners could then draw samples from the replay memory for learning. The learner also serves as a parameter server to the actors which periodically update their policy parameters from the learner.

In our experiments, we always have access to expert demonstrations. We, therefore adopt the practice from DQfD and DDPGfD and put the demonstrations into our replay buffers. For more details see Algorithms 1, 2.

---

**Algorithm 1** Actor

**Given:**
- an experience replay memory $\mathcal{M}$

**for** $n_{episodes}$ **do**
  **for** $t = 1$ **to** $T$ **do**
    Sample action from task policy: $a_t \leftarrow \pi(s_t)$
    Execute action $a_t$ and observe new state $s_{t+1}$, and reward $r_t$.
    Store transition $(s_t, a_t, r_t, s_{t+1})$ in memory $\mathcal{M}$
  **end for**
**end for**

---

**Algorithm 2** Learner

**Given:**
- an off-policy RL algorithm $\mathbb{A}$
- a replay buffer $\mathcal{M}$
- a replay buffer of expert demonstrations $\mathcal{M}_e$

Initialize $\mathbb{A}$
**for** $n_{updates}$ **do**
  Sample transitions $(s_t, a_t, r_t, s_{t+1})$ from $\mathcal{M}$ to make a minibatch $B$.
  Sample transitions $(s_t, a_t, r_t, s_{t+1})$ from $\mathcal{M}_e$ enlarge the minibatch $B$.
  Perform a actor update step with Eqn. (5).
  Perform a critic update step with Eqn. (6).
  Update the target actor/critic networks every $k$ steps.
**end for**

---

## A.7 NETWORK ARCHITECTURE AND HYPERPARAMETERS

Actor and critic share a residual pixel encoder network with eight convolutional layers (3x3 convolutions, three 2-layer blocks with 16, 32, 32 channels), instance normalization (Ulyanov et al., 2016) and exponential linear units (Clevert et al., 2015) between layers.

The policy is a 3-layer MLP with ReLU activations with hidden layer sizes (300, 200). The critic is a 3-layer MLP with ReLU activations with hidden layer sizes (400, 300). For a illustration of the network. Please see Figure 15.

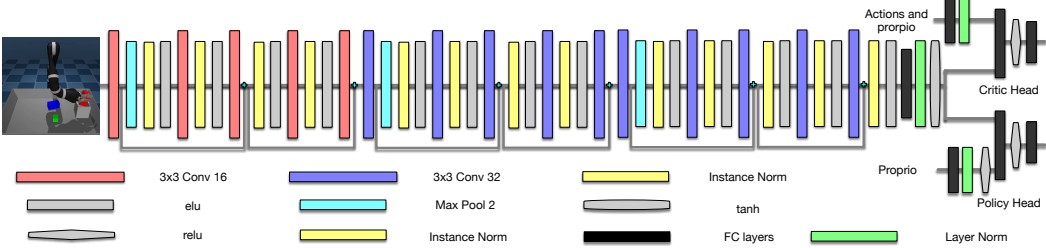

Figure 15: Network architecture for the policy and critic.

The discriminator network uses a pixel encoder of the same architecture as the actor critic, followed by a 3-layer MLP with ReLU activations and hidden layer sizes (32, 32).

| Parameters | Values |
|---|---|
| Actor/Critic Input Width | 64 |
| Actor/Critic Input Height | 64 |
| **D4PG Parameters** | |
| $V_{min}$ | $-50$ |
| $V_{max}$ | 150 |
| $V_{bins}$ | 21 |
| N step | 1 |
| Actor learning rate | $10^{-4}$ |
| Critic learning rate | $10^{-4}$ |
| Optimizer | Adam (Kingma & Ba (2014)) |
| Batch size | 256 |
| Target update period | 100 |
| Discount factor ($\gamma$) | 0.99 |
| Replay capacity | $10^6$ |
| Number of actors | 32 or 128 |
| **Imitation Parameters** | |
| Discriminator learning rate | $10^{-4}$ |
| Discriminator Input Width | 48 |
| Discriminator Input Height | 48 |

Table 3: Hyper parameters used in robot manipulation experiments.

## A.8 COMPARING AGAINST LEARNING WITH FIXED REWARDS

To check whether learned discriminators are needed, we compare TRAIL against agents using a fixed reward function corresponding to $R_{expert} = 1$ and $R_{agent} = 0$ for the *lift alone* and *lift distracted* tasks. This baseline simulates an oracle discriminator with perfect generalization, but which is agnostic to behavior. On the *lift alone* task, agents using this fixed reward achieve roughly half the

reward of TRAIL asymptotically, and on *lift distracted* they do not solve the task (average rewards are less than 5). The learning curves are presented in Figure 16.

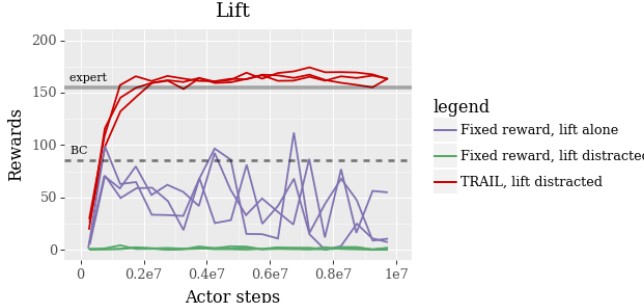

Figure 16: With fixed rewards, the agent is able to learn *lift alone* somewhat, but performs worse than TRAIL. When distractor blocks are added, the fixed reward agent fails to learn completely.

