# OpenReview forum: "Task-Relevant Adversarial Imitation Learning"
_ICLR.cc/2020/Conference — Reject_

### Official Review · AnonReviewer3 · 2019-10-23
**Official Blind Review #3**

**Rating:** 8

**Review:**

The paper proposes an extension of the adversarial imitation learning framework where the discriminator is additionally incentivized not to distinguish frames that are different between the expert and the agent in irrelevant ways. The method relies on manually identifying a set of irrelevant frames.

The paper correctly identifies an important shortcoming of GAIL and proposes a sensible and generic way to overcome it. The experiments are well designed and corroborate the claims made in the paper.

EDIT: I acknowledge reading the other reviews and the author response and stand by my initial assessment.

**Experience Assessment:**

I do not know much about this area.

**Review Assessment: Checking Correctness Of Derivations And Theory:**

I assessed the sensibility of the derivations and theory.

**Review Assessment: Checking Correctness Of Experiments:**

I assessed the sensibility of the experiments.

**Review Assessment: Thoroughness In Paper Reading:**

I made a quick assessment of this paper.

---

> ### Author Response · Authors · 2019-11-06
> **Thanks for the review**
>
> We would like to thank for the review. We found the summary very accurate and identifying the most important take-aways from our work. We are glad that the reviewer assigned the highest possible score and voted for acceptance.

---

### Official Review · AnonReviewer4 · 2019-10-28
**Official Blind Review #4**

**Rating:** 3

**Review:**

The paper addresses the acute problem of irrelevant discrimination in adversarial imitation learning methods. Ideally, the discriminator in such methods should extract task-dependent features and base its discrimination rule upon them only. However, in practice, the opposite usually occurs. For example, if the state distributions of the expert and agent come from different modalities, e.g., different background light, different POV camera, etc., then almost surely the discriminator will base its classification rule on such differences. In turn, the reward signal will be (constant and) irrelevant. This is a well-known problem, that indeed received little attention in previous work.
As a solution, the authors propose to maintain an invariance set representing task-irrelevant states. For example, states of a random policy or states from the beginning of an episode. The authors suggest using the invariance set as a regularization mechanism to the discriminator by punishing it for correctly classifying examples from the invariance set. The idea behind this regularization is to distinguish between expert examples that actually represent the task and expert examples that do not. This is achieved by punishing for the discriminator's accuracy over the invariance set.
In addition, the authors present two additional improvements: data augmentation and early actor stopping.
The paper.

1. I very much like the problem the authors choose to address. I believe that this is a dominant problem that received little attention so far. The reason is that so far, GAIL style methods used experts and agents that come from the same domains (same simulation), thus "hiding" such problems. As experiments scale up to real-world setups, this is no longer the case and such problems arise.

2. Overall, I find the experiments section very confusing. The authors propose a formal improvement (the invariance set) which they apply to their agent, as well as two informal improvements (actor early stopping and data augmentation) which they apply selectively for their agent and the baseline methods. As a result, there exist a large number of variations, according to what improvements are applied to what method. Overall, I find it hard to follow the results and understand the contribution of the different improvements.

3. It is not clear if TRAIL and GAIL are using the same architecture and algorithm. Apparently, the authors could have used GAIL's original policy gradient algorithm with their invariance set modification. If an improvement was evident in such a comparison, then it would have been easier for me to understand the significance of the proposed method. Currently, there are too many differences between the baseline and TRAIL to clearly say that the invariance set made the difference.

4. It is not clear why the constraint in Eq. 2 is placed over the sum of the two terms and not on each term separately since each term is defined over a different state distribution. What would you expect to see different if the constraint was placed on the two terms separately?

5. In the same matter, why do you need the 0.5 factor?

6. The invariance set is referred to extensively before it is first defined. Therefore, the reader is forced to toggle back and forth multiple times in section 4 before understanding equations 2 and 3. Consider giving at least an intuition about the invariance set beforehand to make the reading more fluent.

7. The conclusions section is minimal to non-existing while there are many conclusions to draw from the experiments.

8. Data augmentation was previously studied in the context of adversarial imitation. See for example "Visual imitation with reinforcement learning using recurrent Siamese networks" by Berseth et.al.

9. Overall assessment: among the three modifications the authors propose, I would be glad to see if the invariance set modification alone can make a difference. The other two modifications are either not novel (see Berseth et.al.), or highly heuristic ("...we restart an episode if the discriminator score at the current step exceeds the median score of the episode so far for T_patience consecutive steps..."). However, it seems that the informal improvements (augmentation and early stopping) play a crucial role in the success of TRAIL (... It drastically improves the baseline GAIL agent, and is necessary to solve any of the harder tasks..."). It is absolutely fine that the authors stress the importance of the informal improvements but it seems that they are dominating the contribution of the main ingredient.

I also find it problematic that the construction of the invariance set is not straightforward ("...the selection of the invariance set is a design choice..."). When added to the way that the data augmentation is carried (sensor dependent), and to the fact the the early stopping mechanism seems dubious for some tasks (I can think of several tasks where early stopping will not work), this method is far from being a "plug-and-play" solution. I.e., the paper offers general guidelines on how to solve imitation tasks but not a concise algorithm.
In order for me to vote for acceptance, I need the paper to be more concise. In its current state, I see it more as general guidelines for training adversarial imitation algorithms.

However, as said in the beginning, I very much believe in the problem that the authors present and encourage them to provide an experiment where the only modification between the baseline and the proposed method is the usage of an invariance set. Success in such an experiment will clearly change my mind.

There are also some presentation issues, e.g cluttered graphs, that can be addressed.

**Experience Assessment:**

I have published one or two papers in this area.

**Review Assessment: Checking Correctness Of Derivations And Theory:**

N/A

**Review Assessment: Checking Correctness Of Experiments:**

I did not assess the experiments.

**Review Assessment: Thoroughness In Paper Reading:**

I read the paper at least twice and used my best judgement in assessing the paper.

---

> ### Author Response · Authors · 2019-11-06
> **Significance of using the invariant set constraint**
>
> We would like to sincerely thank R4 for the thorough review. In this comment we address the main concern of R4: whether use of the invariant set constraint significantly contributes to the improved performance of TRAIL over GAIL. We present evidence that it does contribute decisively, addressing reviewer comments numbered 2, 3 and 9. The other reviewer comments are answered in a separate response.
>
> To isolate the effect of using invariant set constraints in TRAIL, see for example figure 8. All three methods (GAIL, TRAIL-0 and TRAIL) use data augmentation. TRAIL-0 and TRAIL use adaptive early stopping, but only TRAIL uses invariant set constraints. TRAIL clearly performs best across all tasks. Code, agent configurations (e.g. number of actors, etc) and network architecture are otherwise exactly the same. We also observe consistent results in the varied block lifting experiments described in figure 3.
>
> In all our experiments, baseline GAIL is the strongest GAIL we were able to implement, and also benefits from data augmentation. We will make these details more clear in the revised paper. All common hyper-parameters were tuned for baseline GAIL for the lift-alone task in the early stage of the project. We did not try to tune them separately for TRAIL; our method already solved all tasks considered so tuning was not needed. The hyperparams making baseline GAIL achieve results comparable with BC on lift alone were used for all tasks and discriminator-based methods (GAIL, TRAIL-0 and TRAIL).
>
> We also provide ablations isolating the effect of data augmentation and adaptive early stopping (see Section 5.2), which indicate that these are necessary but not sufficient for solving the full task suite. To solve all the tasks, invariant set constraints were also needed.
>
> We will revise the text to make it clear that the informal improvements - augmentation and adaptive early stopping - were not applied selectively, but rather uniformly except as indicated in the relevant ablations.
>
> We hope that our response has addressed your concern about the significance of the invariant set constraint, but if there is still a missing experiment needed to raise your score, please let us know and we may be able to run it.

---

> ### Author Response · Authors · 2019-11-06
> **Response to numbered comments 1, 4, 5, 6, 7, 8**
>
> Ad 1.
> We are glad that you agree that the problem dealt with is very important. We believe that even only presenting the problem backed by a set of experiments would be valuable, but we also aimed to go a step further and develop a widely applicable solution in the same work.
> Ad 4.
> Our formulation is less strict and allows the discriminator to find solutions that work better as a reward function. We noticed for example that our discriminator learns to assign low scores to all states from the invariant set. It means that the discriminator behaves more like a reward model assigning higher values only if an input observation manifests at least a partially solved task. When the two constraint terms are enforced separately, the discriminator is forced to predict higher scores near 0.5 for invariant set states, which we found to perform worse as a reward function, perhaps due to decreased modeling flexibility.
> Ad 5.
> When the 0.5 threshold is not used, the algorithm is much more sensitive to the $\lambda$ hyperparameter because regularization gradient flipping would be always applied. Intuitively, this value (0.5) corresponds to a discriminator that is no better than making a random guess on the invariant set. Practically, the effect is for the final algorithm to be less sensitive to $\lambda$.
> Ad 6., Ad 7.
> Thank you for the feedback; we will give a better intuition about the invariant set before using the term, and expand the conclusion section.
> Ad 8.
> We will add this reference in the revised version.
>
> If there is any question you would like to ask, we are happy to discuss.

---

### Official Review · AnonReviewer1 · 2019-10-30
**Official Blind Review #1**

**Rating:** 6

**Review:**

UPDATE:
I acknowledge that I‘ve read the author responses as well as other reviews.
I would keep my score unchanged, i.e. at 6 Weak Accept.


#####

This work considers imitation learning, i.e. the problem of learning an unknown reward function from expert demonstrations. The paper proposes Task-Relevant Adversarial Imitation Learning (TRAIL), a modification of Generative Adversarial Imitation Learning (GAIL) (Ho & Ermon, 2016) which introduces an additional constraint to the discriminator optimization, namely that the discriminator must not be able to successfully discriminate between expert and agent samples on some predetermined invariant set $\mathcal{I}$. Technically this constraint is realized with an additional term added to the standard GAIL objective that minimizes the negative (i.e. reversed) discriminator objective whenever discrimination accuracy exceeds 50% (i.e. randomness) on $\mathcal{I}$. Given that the invariant set $\mathcal{I}$ includes environment samples not relevant for the task, TRAIL thus aims to prevent discriminator overfitting to task-irrelevant features (e.g. object or agent appearance) and regularizes the discriminator to extract more salient, task-relevant features of expert behavior.
In experiments on various robot manipulation tasks (Block lifting/stacking/insertion with and without distraction), the paper shows that TRAIL outperforms Behavioral Cloning (BC), standard GAIL, Deterministic Policy Gradients from Demonstrations (DPGfD) as well as additionally introduced improved GAIL baselines that employ common regularization techniques (early stopping and data augmentation).

I think this work could be accepted since it clearly demonstrates a key issue of GAIL (tendency to overfit to task-irrelevant features) and proposes a simple solution for this issue that empirically proves to be effective. I especially appreciate the comprehensive ablation studies that isolate effects from data augmentation and actor early stopping. The paper overall is well written and easy to follow.
I must admit, however, that I am not too familiar with this line of research and might be missing some context, e.g. recent competitors.

Some questions that I have:
(1) Could you elaborate more on how to select the invariant set $\mathcal{I}$ in practice? The paper proposes random policies or initial frames, but as I understand experiments are only reported for the latter design choice. How would performance differ for different choices? What if the invariant set $\mathcal{I}$ is misspecified, i.e. includes task-relevant samples? Would TRAIL be robust?
(2) How is the invariant set $\mathcal{I}$ chosen for the stack, insertion, and stack banana experiments in Figure 8?
(3) What if actions would be included in objective (2) again?


#####
Minor comments
- Is the legend in Figure 14 wrong? Table 1 results and the plots in Figure 14 are inconsistent (e.g. on Lift distracted "TRAIL" is best in Figure 14 which should be "TRAIL - augmentation" according to Table 1.)
- Section 5.1: "Finally, to disentangle the importance of actor early stopping, ..." >> "disentangle" somewhat is a signal word in ML that readers directly associate with representation learning. Maybe just "show"/"infer"/"understand".


**Experience Assessment:**

I do not know much about this area.

**Review Assessment: Checking Correctness Of Derivations And Theory:**

N/A

**Review Assessment: Checking Correctness Of Experiments:**

I assessed the sensibility of the experiments.

**Review Assessment: Thoroughness In Paper Reading:**

I read the paper at least twice and used my best judgement in assessing the paper.

---

> ### Author Response · Authors · 2019-11-06
> **Response to questions 1, 2, and 3 and minor comments**
>
> We thank R1 for your valuable feedback. Below we respond to your questions:
>
> (1) In practice, we found it generally applicable to use early frames. However, we did also try using random agent frames in a later experiment, and the resulting agent was able to solve the lifting task as well as the agent using early frames for the invariant set. Figure 4 shows a comparison between these two strategies for one of the hardest tasks considered — both methods work comparable and clearly outperform GAIL.
>
> We expect that the performance of a particular choice of invariant set would be best when it covers a diverse space of behaviors that are not task relevant. If not enough varied behavior is shown in the invariant set, it probably will not result in the discriminator learning a suitable reward function. If an invariant set is mis-specified, or contains observations that are in fact task-relevant, then we expect reward functions would be impaired resulting in worse task performance.
>
> (2) In figure 8 we used early frames, configured to the default of using the first 10 across all tasks.
>
> (3) We found including actions to be harmful to performance, allowing the discriminator to distinguish agent and expert to easily. This may be due to the difficulty of exactly reproducing the distribution of expert actions, which is actually not needed to get a performant agent.
>
> We want to note that we use human expert demonstration which is known to be much more challenging. In the context of the question, it makes the expert and agent action easier to discriminate. In our case, for example, due to teleoperation nature of the expert data, gripper-related values are always -1, 1 or 0 for expert (meaning close, open or do not change), while they are continuous for agent data.
>
> Minor comments:
> - By “TRAIL - augmentation” we meant “TRAIL minus augmentation”. Thank you for pointing this out, we will fix this labeling which was indeed confusing.
> - We will change “disentangle” to a more suitable word like “show”.

---

### Author Response · Authors · 2019-09-27
**1-minute video showing GAIL and TRAIL agents performing tasks considered in this work**

We would like to share a video showing GAIL and TRAIL agents performing tasks considered in this work: https://youtu.be/Rz5G15rDKcg .

---

### Public Comment · ~Super_User1 · 2019-10-18
**Paper modification date updated.**

This paper's author list was modified to remove a typo.

Best,
OpenReview Team

---

### Author Response · Authors · 2019-11-15
**The revision uploaded**

We uploaded the revised version of the paper. We mainly focused on changes discussed already in our rebuttal. To list the most important:
- We made clear that all discriminator-based methods in this work (including all baselines) use data augmentation except as indicated in the relevant ablations.
- We explained that the only difference between TRAIL-0 and TRAIL is that the later uses invariant set constraints, and hence the decisive contribution of the latter is clear.
- We expanded the conclusion section.
- We gave a better intuition about the invariant set before using the term.
- We addressed minor remarks raised by reviewers (corrected labels in Figure 14, changed 'disentangle' word, added reference to Berseth et al. 2019).

---

### Decision · Program_Chairs · 2019-12-19

**Decision:**

Reject

**Comment:**

This paper attempts to improve adversarial imitation learning (GAIL) by encouraging the discriminator to focus on task-dependent features.

An advantage of this paper is that it not only improves upon GAIL, but it is doing so after first demonstrating and analyzing an existing issue.

On the other hand, the presentation of the paper and breadth of experiments could be significantly improved further than the updated version. It would also be necessary to clarify whether the baseline is vanilla PG or D4PG.

A major point for discussion was the selection of the invariance set. The ablation studies and explanation provided during the rebuttal period towards this point are helpful, but somehow we still do not have the full picture to understand well how this method compares to existing literature.